# Secure Communication in Cooperative SWIPT NOMA Systems with Non-Linear Energy Harvesting and Friendly Jamming

**DOI:** 10.3390/s20041047

**Published:** 2020-02-14

**Authors:** Van Phu Tuan, Ic-Pyo Hong

**Affiliations:** Department of Information & Communication Engineering, Kongju National University, 1223-24 Cheonan-dae-ro, Cheonan 31080, Korea; phutuan87@gmail.com

**Keywords:** physical layer security, non-linear energy harvesting, non-orthogonal multiple-access (NOMA), cooperative communication, friendly jamming

## Abstract

This paper studies the secure communication of a non-orthogonal multiple-access (NOMA) relaying system in the presence of an eavesdropper in which the NOMA communication between a source and two users is assisted by an energy-harvesting (EH) relay. The relay extracts a part of its received signal strength using a power-splitting (PS) policy then harvests energy using a non-linear EH (NLEH) circuit. A friendly jammer sends jamming signals to help secure communication. The jammer is exploited as an additional energy source. A store-and-transmit (SaT) scheme which allows the EH relay to perform energy storing and information transmitting is proposed. For performance evaluation, the closed-form expressions for three metrics, secrecy outage probability (SOP), average achievable secrecy rate (AASR) and average stored energy (ASE) are derived. These results enable studies on the effects of various system parameters, such as NOMA power-allocation factors, target secrecy rates, jammer’s location, and relay’s power levels, on the system performance.

## 1. Introduction

The rapidly growing demand for radio connectivity in both industrial and residential use has caused various issues in wireless networks: faster battery drain at the device, various access required at the base station, and higher security risk. This motivated studies on relevant efficient techniques. Wireless energy harvesting (WEH), which enables devices charging by scavenging energy from ambient radio signals, physical layer security (PLS), which provides secure communication by exploiting the physical nature of wireless channels, and non-orthogonal multiple access (NOMA), which supports high spectral efficiency, low latency, and massive access by modulating signal on the power domain, are expected as strong candidates to help the future wireless systems overcome these issues.

Based on the fact that radio signal can carry both the energy and information, the author of [1] designed two WEH receiver architectures to allow a simultaneous wireless information and power transfer (SWIPT), time-switching (TS) and power-splitting (PS) where the energy harvesting (EH) and information decoding (ID) functions are separated in time domain and power domain, respectively. The trade-off between harvested energy and information rate of diverse SWIPT systems using TS and PS policies was characterized in [1,2,3]. The works of [4,5] examined different techniques to improve the performance of SWIPT systems via enhancing the amount of harvested energy, such as multi-antenna energy harvester, multi-band energy harvester, and interference-aided WEH. In addition, the work of [6] proposed to exploit both downlink and uplink transmissions of a multi-user TMDA-based system to boost the harvested energy at users. The presence of WEH receivers can be regarded as a potential security threat to wireless communication; hence, security in SWIPT has gained much interest from recent studies. The works of [7,8] used a beamforming technique to increase the secrecy capacity and/or harvested energy of multi-antenna SWIPT systems by optimizing the beamforming vectors. On the other studies, friendly jamming signals were employed to improve secure communication via an EH relay in which the jammer can interfere with wire-tapped information and provide additional energy at the relay [9]. Another approach presented in [10,11] is to use WEH nodes as jammers that employ their harvested energy to assist the secure links.

In the NOMA scenario, the work of [12] designed strategies for a wireless-powered uplink NOMA to maximize the system throughput or achieve the high fairness in throughput of users. A similar work concerning the energy-and-rate efficiency was presented in [13]. The authors of [14] exploited the advantage in distance of the near user (NU) to perform a SWIPT-based cooperative communication; hence, NU can set up an additional link to the far user (FU) using its harvested energy. The work of [15] designed a strategy to maximize the harvested energy of each harvester in a cognitive radio (CR) NOMA system, especially, the non-linear energy harvesting (NLEH) model was considered. The works of [16,17] investigated the NOMA communications for the WEH machine-to-machine networks and focused on minimizing the total energy consumption at the devices and gateways via joint power control and time allocation. To guarantee security in multi-user networks, the studies on secure communication of SWIPT-based NOMA systems are needed. The work of [18] investigated the secure transmission of a WEH relaying networks via analyzing the secrecy outage probability (SOP) and secrecy throughput in a delay-limited mode. The works of [19] considered a downlink NOMA system containing several EH receivers that can wiretap confidential information. To maximize the sum secrecy rate of this system, a strategy for optimizing NOMA power-allocation factors was proposed. The authors of [20] studied secrecy energy efficient in a CR-NOMA system with NLEH. This work focused on developing strategies to find the optimal transmit powers of WEH secondary transmitters for given conditions of energy channels.

It is seen that most studies on PLS in SWIPT-based NOMA systems were performed under the assumption of a linear energy harvesting (LEH) model. In addition, the NLEH model proposed in [21] is not mathematically tractable [22]; hence, simpler alternative NLEH models are employed to obtain analytical results for system performance. Motivated by the above assessments, this paper studies the secure communication of a SWIPT-based cooperative NOMA system with NLEH and friendly jamming, and derives analytical results that facilitate performance evaluation and system design. In this system, a source communicates with two users via a NLEH relay that harvests energy form the source signal and jamming signal. The use of friendly jammer can reduce the security risk from a passive eavesdropper. The contributions of this paper are described as follows.We propose a store-and-transmit (SaT) scheme providing multi-transmit-power levels (MTPLs) at the EH relay. This simplifies the design of EH transceivers, reduces the energy wasted due to ineffective connectivity, and enables the relay to store a part of energy to maintain its important activities (for the harvest-to-transmit (HtT) scheme, the EH relay uses all the harvested energy to transmit information and maintains its important activities with its limited battery reserves). Moreover, the SaT scheme allows the calculations in performance to become analytically tractable. By using accordant MTPLs, the obtained analytical results for the SaT scheme can allow evaluations in the system performance of the HtT scheme.Three performance metrics, SOP, average achievable secrecy rate (AASR) and average stored energy (ASE), are studied for performance evaluation. We derive closed-form expressions for the SOP (for the whole communication and for the case that the relay is active) and the AASR of each user; in addition, we derive an exact analytical expression for the ASE. We use Montes-Carlo simulations to verify the accuracy of the analytical results.Finally, the effect of various key system parameters, the location of jammer, target secrecy rates, NOMA power-allocation factors, transmit-power strategies (TPSs) at the EH relay, on the system performance, is studied to provide insight into the system design.

The rest of this paper is organized as follows. The system model and preliminary results are presented in Section 2. In Section 3, analytical expressions for the SOP, AASR and ASE are derived. Simulation results and discussions are presented in Section 4. Finally, the conclusion is presented in Section 5. Appendix A–Appendix D present the proofs of the propositions.

*Notation*: We use A→B to denote the link from node *A* to node *B*; ⊗ is the XOR operator; $\mathrm{CN}(0,N_{0})$ is a complex Gaussian distribution with zero mean and variance $N_{0}$; Ei(·) is the exponential integral function [23] (Eq. (8.211.1)); ${}_{p}F_{q}\left(-;-;-\right)$ is the generalized hypergeometric function [23] (Eq. (9.14.1)); E{X} is the expectation of a random variable (RV) *X*; E{X|Y} is the conditional expectation of *X* given an event *Y*; and ${\left[X\right]}^{+}=\max\left\{X,0\right\}$.

## 2. System Model and Preliminary Results

### 2.1. System Model

We consider an EH relaying NOMA system, as shown in Figure 1a, including a source *S*, a destination *D*, a decode-and-forward (DF) NLEH relay *R*, a friendly jammer *J*, an eavesdropper *E* and two users, i.e., a NU $U_{1}$ and a FU $U_{2}$. Due to the presence of *E*, which tries to overhear the user’s confidential information, *J* is employed to send jamming signals to interfere with *E* and these signals can be exploited as an additional energy source for *R*. All nodes are single–antenna devices. We assume that (i) $U_{1}$, $U_{2}$ and *E* are close to the cell edge; hence, there are no direct links from *S* to $U_{1},U_{2}$ and *E*, and the users and *E* decode the source’s information via relay’s forwarded signal; (ii) the channels undergo independently and non-identically distributed (i.n.i.d.) Rayleigh fading; (iii) the channels are constants during a block time *T*; (iv) local channel state information (CSI) is available at each node; and (v) the jamming signal is only revealed to legitimate nodes.

Assumption (v) is explained as the follows. Each legitimate node $U\in \{R,U_{1},U_{2}\}$ and *J* establish a shared secret key, i.e., $K_{\left(U-J\right)}$, using CSI-based key generation method [24]. Then *J* broadcasts two combined keys $K_{\left(R-J\right)}⊗K_{\left(U_{1}-J\right)}$ and $K_{\left(R-J\right)}⊗K_{\left(U_{2}-J\right)}$. Using their own key, all legitimate nodes can discover all secret keys while *E* cannot do that due to lack of global CSI; hence, only legitimate nodes can eliminate the jamming signal on their received signals. Let $\tau_{\mathrm{setup}}$ denote the duration for the setup phase consuming for channel estimation and secret key generation. The remain of block time *T* is separated into two equal time slots, $\tau_{0}=\left(T-\tau_{\mathrm{setup}}\right)/2$, (illustrated in Figure 1b) and the communication during each time slot is described in Section 2.3 and Section 2.4.

### 2.2. Non-Liner Energy Harvesting Model and Store-and-Transmit Protocol

Figure 2a–c show the relations between the received power $P_{\mathrm{Rx}}$ and harvested power $P_{{}_{\mathrm{EH}}}$ for the LEH and NLEH models. The LEH model assumes a linear relationship $P_{\mathrm{EH}}=\eta P_{\mathrm{Rx}}$ at the harvester where η is energy conversion efficiency. However, measurements in [25,26] show the non-linear relationship between $P_{\mathrm{Rx}}$ and $P_{\mathrm{EH}}$ which is well modeled using a logistic function as [18]
(1)$$P_{\mathrm{EH}}=\Phi_{\mathrm{EH}}\left(P_{\mathrm{Rx}}\right)=P_{\mathrm{EH}}^{\max}\left(1-\frac{1+e^{a_{\mathrm{EH}}b_{\mathrm{EH}}}}{e^{a_{\mathrm{EH}}P_{\mathrm{Rx}}}+e^{a_{\mathrm{EH}}b_{\mathrm{EH}}}}\right),$$
where $P_{\mathrm{EH}}^{\max}$ is the maximum harvested power of the harvester, and $a_{\mathrm{EH}}$ and $b_{\mathrm{EH}}$ are EH parameters of the harvester. It is seen from (Equation 1) that the random variables in $P_{\mathrm{Rx}}$ are input variables of an exponential function, this leads to the impossibility of grouping or separating independent random variables to perform further calculations. For that reason, the studies on SWIPT systems commonly adopt the LEH model to simplify calculations instead of using (Equation 1).

In this paper, we propose a new SaT scheme aiming to support the SWIPT relaying systems. Specifically, this scheme allows the EH relay to choose the appropriate transmit-power level and then perform the energy-storing and information-forwarding tasks simultaneously (as illustrated in Figure 2d). The SaT scheme includes MTPLs represented by an output-power space (OPS) $O_{{}_{R,\mathrm{OPS}}}≜\left\{0,P_{{}_{R,1}},\ldots ,P_{{}_{R,N}},P_{\mathrm{EH}}^{\max}\right\}$ where $P_{{}_{R,n}},n=1,\ldots ,N,$ are transmit-power levels supported at *R* and satisfy $0<P_{{}_{R,1}}<\ldots <P_{{}_{R,n}}<\ldots <P_{{}_{R,N}}<P_{\mathrm{EH}}^{\max}$. For a given $O_{{}_{R,\mathrm{OPS}}}$, the instantaneous transmit power of *R*, $P_{{}_{R}}$, is a function of $P_{\mathrm{Rx}}$ determined by a TPS as the follows.
(2)$$\begin{array}{cc} P_{{}_{R}} & =G_{{}_{R}}\left(\Phi_{\mathrm{EH}}\left(P_{\mathrm{Rx}}\right)\right)≜\left\{\begin{array}{cc} P_{{}_{R,N}} & ;\;\mathrm{If}\;P_{{}_{R,N}}\leq \Phi_{\mathrm{EH}}\left(P_{\mathrm{Rx}}\right)<P_{\mathrm{EH}}^{\max}\\ P_{{}_{R,n}} & ;\;\mathrm{If}\;P_{{}_{R,n}}\leq \Phi_{\mathrm{EH}}\left(P_{\mathrm{Rx}}\right)<P_{{}_{R,n+1}}\\ \; & \;\;\;\;,n=1,\ldots ,N-1\\ 0 & ;\;\mathrm{If}\;\Phi_{\mathrm{EH}}\left(P_{\mathrm{Rx}}\right)<P_{{}_{R,1}} \end{array}\right., \end{array}$$

The remaining harvested energy after sending information at power $P_{{}_{R}}$ is stored in the battery of *R* for maintaining other important activities. The power for energy storing is given by
(3)$$P_{{}_{\mathrm{SE}}}=P_{{}_{\mathrm{EH}}}-P_{{}_{R}}=\Phi_{\mathrm{EH}}\left(P_{\mathrm{Rx}}\right)-G_{R}\left(\Phi_{\mathrm{EH}}\left(P_{\mathrm{Rx}}\right)\right).$$

The randomness of $P_{\mathrm{Rx}}$ effects on both $P_{{}_{\mathrm{SE}}}$ and $P_{{}_{R}}$. The ASE is discussed in Section 3.3 and the cumulative distribution function (CDF) of $P_{{}_{R}}$ is presented in Proposition 1. By using the SaT scheme, the calculations for system performance of NLEH relaying systems become analytically tractable. Moreover, with sufficient number of transmits-power levels and a small *“power gap”* between two consecutive transmit-power levels in $O_{{}_{R,\mathrm{OPS}}}$, the system performance for the SaT scheme is close to that for the HtT scheme; hence, the SaT scheme can be considered to be an approximate solution to access the system performance of the HtT-based NLEH relaying systems.

### 2.3. Communication in the First Time Slot

During this time slot, *S* broadcasts an information $x_{{}_{S}}=\sqrt{a_{1}}x_{1}+\sqrt{a_{2}}x_{2}$ at power $E\{|x_{s}{|}^{2}\}=P_{{}_{S}}$ where $x_{1}$ and $x_{2}$ are the desired information of $U_{1}$ and $U_{2}$, respectively, $a_{1}$ and $a_{2}$ are NOMA power-allocation factors at *S* and $a_{1}+a_{2}=1$. *R* uses PS policy, as shown in Figure 1b, to harvest energy. Moreover, *J* gives an addition energy at *R* by sending jamming signal $x_{{}_{J}}$ at power $E\{|x_{{}_{J}}{|}^{2}\}=P_{{}_{J}}$. Letting $h_{{}_{AB}}$ and $d_{{}_{AB}}$ be the channel coefficient and distance of the A→B link (node *A* and node *B* can be $S,R,U_{1},U_{2},E$ and *J*), respectively, the received information signal at the antenna of *R* is $y_{{}_{R}}=h_{{}_{SR}}x_{{}_{S}}+h_{{}_{JR}}x_{{}_{J}}$. Next, *R* uses a portion $\sqrt{1-\theta }y_{{}_{R}}$ for EH and the rest portion $\sqrt{\theta }y_{{}_{R}}$ for information decoding (ID); hence, the received power at the EH component and input signal at the ID component are expressed as
(4)$$\begin{array}{cc} P_{\mathrm{Rx}} & =\left(1-\theta \right)\left(g_{{}_{SR}}P_{{}_{S}}+g_{{}_{JR}}P_{{}_{J}}\right), \end{array}$$
(5)$$\begin{array}{cc} y_{{}_{R}}^{\mathrm{ID}} & =\sqrt{\theta }h_{{}_{SR}}x_{{}_{S}}+\sqrt{\theta }h_{{}_{JR}}x_{{}_{J}}+n_{{}_{R}}, \end{array}$$
where $g_{{}_{AB}}={\left|h_{{}_{AB}}\right|}^{2}$; and $n_{{}_{R}}∼\mathrm{CN}\left(0,N_{0}\right)$ is an additive white Gaussian noise (AWGN) at *R*. From Assumption (ii), $g_{{}_{AB}}$ is an exponential RV with parameter $\lambda_{{}_{AB}}=d_{{}_{AB}}^{-\alpha }$ where α is path loss exponent. The probability density function (PDF) and CDF of $g_{{}_{AB}}$ are given by $f_{g_{{}_{AB}}}\left(x\right)=\lambda_{{}_{AB}}e^{-\lambda_{{}_{AB}}x}$ and $F_{g_{{}_{AB}}}\left(x\right)=1-e^{-\lambda_{{}_{AB}}x}$, respectively.

Since *R* can eliminate the effect of the jamming signal in (5), the SNR for $x_{1}$ and the SINR for $x_{2}$ obtained at *R* by using successive interference cancellation (SIC) receiver are given by
(6)$$\begin{array}{cc} \gamma_{R}^{x_{1}} & =\mu_{1}g_{{}_{SR}}, \end{array}$$
(7)$$\begin{array}{cc} \gamma_{R}^{x_{2}} & =\frac{\mu_{2}g_{{}_{SR}}}{\mu_{1}g_{{}_{SR}}+1}\overset{\left(a\right)}{\approx }\frac{a_{2}}{a_{1}}, \end{array}$$
where the parameters $\mu_{k},1⩽k⩽11$, are defined in Table 1. The approximation at (*a*) is based on the assumption that the noise power is small as compared to the interference caused by $x_{1}$.

**Proposition** **1.**
*The CDF and PDF of a RV $X≜\frac{P_{\mathrm{Rx}}}{1-\theta }=\left(g_{{}_{SR}}P_{{}_{S}}+g_{{}_{JR}}P_{{}_{J}}\right)$ are respectively expressed by*
(8)$$\begin{array}{cc} \; & F_{X}\left(x\right)=\left\{\begin{array}{cc} 1+\frac{\lambda_{{}_{JR}}P_{{}_{S}}}{\mu_{3}}e^{-\frac{\lambda_{{}_{SR}}}{P_{S}}x}-\frac{\lambda_{{}_{SR}}P_{{}_{J}}}{\mu_{3}}e^{-\frac{\lambda_{{}_{JR}}}{P_{J}}x}\; & ;\mathrm{If}\mu_{3}\neq 0\\ 1-\left(1+\frac{\lambda_{{}_{SR}}x}{P_{S}}\right)e^{-\frac{\lambda_{{}_{SR}}}{P_{S}}x} & ;\mathrm{If}\mu_{3}=0 \end{array}\right., \end{array}$$
(9)$$\begin{array}{cc} \; & f_{{}_{X}}\left(x\right)=\left\{\begin{array}{cc} \frac{\lambda_{{}_{SR}}\lambda_{{}_{JR}}}{\mu_{3}}\left(e^{-\frac{\lambda_{{}_{JR}}}{P_{J}}x}-e^{-\frac{\lambda_{{}_{SR}}}{P_{S}}x}\right) & ;\mathrm{If}\mu_{3}\neq 0\\ \frac{\lambda_{{}_{SR}}^{2}}{P_{S}^{2}}xe^{-\frac{\lambda_{{}_{SR}}}{P_{S}}x} & ;\mathrm{If}\mu_{3}=0 \end{array}\right.. \end{array}$$

*The CDF of $P_{{}_{R}}$, $F_{P_{R}}\left(Y\right),$ where $Y\in O_{{}_{R,\mathrm{OPS}}}$ and the probability $\mathrm{Pr}\left(P_{{}_{R}}=P_{{}_{R,n}}\right),1\leq n\leq N,$ are respectively expressed by*
(10)$$\begin{array}{cc} \; & F_{{}_{P_{R}}}\left(Y\right)=\mathrm{Pr}\left(P_{{}_{R}}<Y\right)=F_{{}_{X}}\left(P(Y)\right), \end{array}$$
(11)$$\begin{array}{cc} \; & \mathrm{Pr}\left(P_{{}_{R}}=P_{{}_{R,n}}\right)=F_{{}_{X}}\left(P_{n+1}\right)-F_{{}_{X}}\left(P_{n}\right), \end{array}$$
*where $P\left(Y\right)=\frac{1}{a_{\mathrm{EH}}\left(1-\theta \right)}\ln\left(\frac{P_{\mathrm{EH}}^{\max}\left(e^{a_{\mathrm{EH}}b_{\mathrm{EH}}}+1\right)}{P_{\mathrm{EH}}^{\max}-Y}-e^{a_{\mathrm{EH}}b_{\mathrm{EH}}}\right),P_{n}=P\left(P_{{}_{R,n}}\right)$ and $P_{N+1}=P\left(P_{\mathrm{EH}}^{\max}\right)=\infty$.*


**Proof.** (Equation 8) is obtained via solving $F_{{}_{X}}\left(x\right)=\mathrm{Pr}\left(g_{0}\leq \frac{x-g_{{}_{JR}}P_{J}}{P_{S}}\right)$. Next, (9) is obtained by taking the derivative of (Equation 8). Finally, using (Equation 1) and (Equation 8), (Equation 10) and (11) can be proven. □

### 2.4. Communication in the Second Time Slot

After decoding $x_{1}$ and $x_{2}$, *R* encodes them as $x_{{}_{R}}=\sqrt{b_{1}}x_{1}+\sqrt{b_{2}}x_{2}$ then forwards $x_{{}_{R}}$ to the users at power $E\{|x_{{}_{R}}{|}^{2}\}=P_{{}_{R}}$ where $b_{1}$ and $b_{2}$ are NOMA power-allocation factors at *R* and $b_{1}+b_{2}=1$. During this time slot, *J* also sends jamming signal $x_{{}_{J}}$ at power $P_{{}_{J}}$ to interfere with *E*. The received signals at nodes $\left\{U_{1},U_{2},E\right\}$ are respectively given by
(12)$$y_{{}_{\left\{U_{1},U_{2},E\right\}}}=h_{{}_{R\{U_{1},U_{2},E\}}}x_{{}_{R}}+h_{{}_{J\{U_{1},U_{2},E\}}}x_{{}_{J}}+n_{{}_{\left\{U_{1},U_{2},E\right\}}},$$
where $n_{{}_{\left\{U_{1},U_{2},E\right\}}}$ is the AWGN at $\left\{U_{1},U_{2},E\right\}$.

Since $U_{1}$ and $U_{2}$ can eliminate the effect of $x_{{}_{J}}$ on its signal, the SINRs for $x_{2}$ at $U_{1}$ and $U_{2}$ are respectively given as
(13)$$\begin{array}{c} \gamma_{U_{1}}^{x_{2}}=\frac{b_{2}P_{R}g_{{}_{RU_{1}}}}{b_{1}P_{R}g_{{}_{RU_{1}}}+N_{0}}\overset{\left(b\right)}{\approx }\frac{b_{2}}{b_{1}}, \end{array}$$
(14)$$\begin{array}{c} \gamma_{U_{2}}^{x_{2}}=\frac{b_{2}P_{R}g_{{}_{RU_{2}}}}{b_{1}P_{R}g_{{}_{RU_{2}}}+N_{0}}\overset{\left(c\right)}{\approx }\frac{b_{2}}{b_{1}}. \end{array}$$

Using the SIC receiver, $U_{1}$ can decode $x_{1}$ with the SNR given as
(15)$$\gamma_{U_{1}}^{x_{1}}=\frac{b_{1}P_{R}g_{{}_{RU_{1}}}}{N_{0}}.$$

Since *E* cannot eliminate the effect of $x_{{}_{J}}$ on its signal, the SNR for $x_{1}$ and SINR for $x_{2}$ obtained at *E* by using SIC receiver are given as
(16)$$\begin{array}{c} \gamma_{E}^{x_{1}}=\frac{b_{1}P_{R}g_{{}_{RE}}}{g_{{}_{JE}}P_{J}+N_{0}}\overset{\left(d\right)}{\approx }b_{1}\hat{Z}, \end{array}$$
(17)$$\begin{array}{c} \gamma_{E}^{x_{2}}=\frac{b_{2}P_{R}g_{{}_{RE}}}{b_{1}P_{R}g_{{}_{RE}}+P_{J}g_{{}_{JE}}+N_{0}}\overset{\left(e\right)}{\approx }\frac{b_{2}}{b_{1}+{\hat{Z}}^{-1}}, \end{array}$$
where $\hat{Z}=\frac{P_{{}_{R}}g_{{}_{RE}}}{P_{{}_{J}}g_{{}_{JE}}}$. Moreover, we denote $Z=\frac{P_{{}_{R,n}}g_{{}_{RE}}}{P_{{}_{J}}g_{{}_{JE}}}$ as a RV which represents for $\hat{Z}$ as $P_{{}_{R}}=P_{{}_{R,n}}$.

The approximations at (b),(c),(d) and (e) are based on the assumption that the noise power is small as compared to the interference caused by $x_{1}$ and $x_{{}_{J}}$.

**Proposition** **2.**
*The CDF and the PDF of Z are given by*
(18)$$\begin{array}{c} F_{{}_{Z}}\left(z\right)=\frac{z}{z+\mu_{5}}, \end{array}$$
(19)$$\begin{array}{c} f_{{}_{Z}}\left(z\right)=\frac{\mu_{5}}{{\left(z+\mu_{5}\right)}^{2}}. \end{array}$$


**Proof.** (Equation 18) and (19) are obtained via solving $F_{Z}\left(z\right)=\mathrm{Pr}\left(g_{{}_{RE}}<\frac{g_{{}_{JE}}P_{J}}{P_{R,n}}z\right)$ and $f_{Z}\left(z\right)=\frac{\partial }{\partial z}F_{Z}\left(z\right)$, respectively. □

### 2.5. The Achievable Secrecy Rates

By definition, the achievable secrecy rate is the maximum secrecy rate for sending confidential information to a legitimate node while the eavesdropper is unable to decode any information. Letting $C_{\left\{R,U_{1},U_{2},E\right\}}^{x_{m}}=\frac{1}{2}\log_{2}\left(1+\gamma_{\left\{R,U_{1},U_{2},E\right\}}^{x_{m}}\right),m\in \left\{1,2\right\},$ be the achievable rate for $x_{m}$ at the node $\left\{R,U_{1},U_{2},E\right\}$, the achievable secrecy rate for $x_{m}$ for overall system is calculated by.
(20)$$\begin{array}{cc} C_{\sec}^{x_{1}} & =\min\left\{C_{R}^{x_{1}},{\left[C_{U_{1}}^{x_{1}}-C_{E}^{x_{1}}\right]}^{+}\right\}, \end{array}$$
(21)$$\begin{array}{cc} C_{\sec}^{x_{2}} & =\min\left\{C_{R}^{x_{2}},{\left[C^{x_{2}}-C_{E}^{x_{2}}\right]}^{+}\right\}, \end{array}$$
where $C^{x_{2}}=\min\left\{C_{U_{1}}^{x_{2}},C_{U_{2}}^{x_{2}}\right\}$ is the condition for successfully decoding $x_{2}$ at both $U_{1}$ and $U_{2}$, hence, SIC at $U_{1}$ operates correctly.

## 3. System Performance Analysis

### 3.1. The Secrecy Outage Probability (SOP)

For NOMA communications, the SOP is defined as the probability of occurring the secrecy outage event, i.e., the achievable secrecy rate of a user is less than its target secrecy rate, at any user. We study the SOP in two scenarios where the outage events are counted over the relay’s transmission duration (i.e., *R* is "ON"), ${\mathrm{SOP}}_{\mathrm{ON}}$, and the whole communication duration (i.e., for both cases *R* is "ON" and "OFF"), ${\mathrm{SOP}}_{\mathrm{All}}$. With given target secrecy rates for $x_{1},R_{\mathrm{th}}^{x_{1}}$, and $x_{2},R_{\mathrm{th}}^{x_{2}}$, ${\mathrm{SOP}}_{\mathrm{All}}$ and ${\mathrm{SOP}}_{\mathrm{ON}}$ are respectively calculated as
(22)$$\begin{array}{cc} P_{\mathrm{All}} & =\mathrm{Pr}\left(C_{\sec}^{x_{1}}<R_{\mathrm{th}}^{x_{1}},C_{\sec}^{x_{2}}<R_{\mathrm{th}}^{x_{2}}\right)\\ \; & =F_{{}_{X}}\left(P_{1}\right)+\underset{P_{1}}{\underset{︸}{\mathrm{Pr}\left(C_{\sec}^{x_{1}}<R_{\mathrm{th}}^{x_{1}},C_{\sec}^{x_{2}}<R_{\mathrm{th}}^{x_{2}},P_{{}_{R}}>0\right)}}, \end{array}$$
where $F_{{}_{X}}\left(P_{1}\right)$ is the probability of the event that *R* is "OFF" and the expression for $P_{1}$ is presented in Proposition 3. $P_{1}$ represents for the outage event that occurs when *R* forwards the information but the instantaneous secrecy rates at the users do not satisfy the user’s target secrecy rates. $P_{1}$ is counted for whole the communication duration, hence, the expression for ${\mathrm{SOP}}_{\mathrm{ON}}$ is obtained using Bayes’s law as follows:(23)$$\begin{array}{cc} P_{\mathrm{ON}} & =\frac{P_{1}}{1-F_{{}_{X}}\left(P_{1}\right)}, \end{array}$$

**Proposition** **3.***The analytical approximate expression for $P_{1}$ is given by*(24)$$P_{1}\approx \sum_{n=1}^{N}\left(P_{b}\left(n\right)+P_{c}\left(n\right)P_{d}^{\approx }\left(n\right)\right),$$*where $P_{b}\left(n\right),P_{c}\left(n\right)$ and $P_{d}^{\approx }$ are presented in*Table 2.

**Proof.** See Appendix A. □

It is seen from (Equation 22) that $F_{{}_{X}}\left(P_{1}\right)$ is a lower bound for ${\mathrm{SOP}}_{\mathrm{All}}$. When $P_{{}_{R,1}}$ is set at high values, the increase in $F_{{}_{X}}\left(P_{1}\right)$ can result the worse ${\mathrm{SOP}}_{\mathrm{All}}$ and better ${\mathrm{SOP}}_{\mathrm{ON}}$ (note that $F_{{}_{X}}\left(P_{1}\right)$ is an increasing function of $P_{{}_{R,1}}$; for practical scenarios, $P_{{}_{R,1}}$ should be greater than a minimum threshold of the incident power, i.e., $P_{\mathrm{th}}^{\mathrm{act}}$, to activate the EH circuit). An appreciate choice for OPS could be exploited to guarantee different goals of the system. For providing reliable communication, the OPS includes high power levels to reduce ineffective communication caused by low relay transmit powers. For frequent communication, the OPS can exploit low power levels. The jamming signal strengths are also an important factor in the SOP. Specifically, the harvested energy is enhanced, and the overhearing rate is decreased when the jamming signal strengths at *R* and *E* increase, respectively. This leads to improvement in the SOP.

### 3.2. The Average Achievable Secrecy Rate (AASR)

#### 3.2.1. The Average Achievable Secrecy Rate for $x_{1}$

Let $O_{P_{R},n}$ be the event that the transmit power of *R* is $P_{R,n}$, i.e., $P_{{}_{R}}=P_{{}_{R,n}}$, $O_{P_{R}}$ be an event space defined as $O_{P_{R}}=\left\{O_{P_{R},1},\ldots ,O_{P_{R},N}\right\}$, and $O_{\hat{Z}}≜\left\{\hat{Z}<\frac{P_{{}_{R}}g_{{}_{RU_{1}}}}{N_{0}}\right\}=\left\{\frac{g_{{}_{RE}}}{P_{{}_{J}}g_{{}_{JE}}}<\frac{g_{{}_{RU_{1}}}}{N_{0}}\right\}$ be the event of existing secure communication for $x_{1}$ during the second time-slot communication (or $O_{\hat{Z}}≜\left\{C_{E}^{x_{1}}<C_{U_{1}}^{x_{1}}\right\}$). The AASR of $x_{1}$, ${\bar{C}}_{\sec}^{x_{1}}$, is calculated by averaging the random variable $C_{\sec}^{x_{1}}$ for all possible cases of $P_{{}_{R}}$ (represented by $O_{P_{R}}$) and $O_{\hat{Z}}$ (due to the presences of random channel gains, $O_{P_{R},n}$ and $O_{\hat{Z}}$ are random events). Using the law of total expectation, ${\bar{C}}_{\sec}^{x_{1}}$ is calculated as
(25)$${\bar{C}}_{\sec}^{x_{1}}=\frac{2\tau_{0}}{T}E\left\{E\left\{E\left\{C_{\sec}^{x_{1}}|O_{P_{R}},O_{\hat{Z}}\right\}|O_{\left\{P_{R},\hat{Z}\right\}}\right\}\right\}.$$

The factor $\frac{2\tau_{0}}{T}=\frac{T-\tau_{\mathrm{setup}}}{T}$ indicates the effect of the setup phase duration on the AASRs. In the multiple-user scenario, the increase in $\tau_{{}_{\mathrm{setup}}}$ causes a significant influence on the amount of the received confidential data. Using the fact that E{min{x,y}}≤min{E{x},E{y}}, we can obtain an upper bound for ${\bar{C}}_{\sec}^{x_{1}}$ as follows.
(26)$${\bar{C}}_{\sec,\mathrm{up}}^{x_{1}}=\frac{2\tau_{0}}{T}\min\left\{\begin{array}{c} E\left\{E\left\{E\left\{C_{R}^{x_{1}}|O_{P_{R}},O_{\hat{Z}}\right\}|O_{\hat{Z}}\right\}\right\},\\ E\left\{E\left\{E\left\{C_{U_{1}}^{x_{1}}-C_{E}^{x_{1}}|O_{P_{R}},O_{\hat{Z}}\right\}|O_{P_{R}}\right\}\right\} \end{array}\right\}.$$

Since $C_{R}^{x_{1}}$ and $O_{\hat{Z}}$ are independent, and $\left(C_{U_{1}}^{x_{1}}-C_{E}^{x_{1}}\right)$ and $O_{P_{R,n}}$ are independent, (Equation 26) can be expressed as in (Equation 27).

**Proposition** **4.***The analytical expression for ${\bar{C}}_{\sec,\mathrm{up}}^{x_{1}}$ is given by the following equation.*(27)$${\bar{C}}_{\sec,\mathrm{up}}^{x_{1}}=\frac{2\tau_{0}}{T}\min\left\{\mathrm{Pr}\left(O_{\hat{Z}}\right)\sum_{n=1}^{N}C_{a}\left(n\right),\sum_{n=1}^{N}C_{b}\left(n\right)\mathrm{Pr}\left(O_{P_{R,n}}\right)\right\},$$*where $\mathrm{Pr}\left(O_{\hat{Z}}\right),\mathrm{Pr}\left(O_{P_{R,n}}\right)$ and $C_{a}\left(n\right)$ are given in (Equation 28)–(30), and the expressions for $C_{a,1}\left(n\right)$ and $C_{b}\left(n\right)$ are presented in*Table 3.(28)$$\begin{array}{cc} \mathrm{Pr}(O_{\hat{Z}}) & =1+\frac{\lambda_{{}_{RU_{1}}}N_{0}\mu_{5}}{P_{R,n}}e^{\frac{\lambda_{{}_{RU_{1}}}N_{0}\mu_{5}}{P_{R,n}}}Ei\left(-\frac{\lambda_{{}_{RU_{1}}}N_{0}\mu_{5}}{P_{R,n}}\right), \end{array}$$(29)$$\begin{array}{cc} \mathrm{Pr}(O_{P_{R,n}}) & =F_{{}_{X}}\left(P_{n+1}\right)-F_{{}_{X}}\left(P_{n}\right), \end{array}$$(30)$$\begin{array}{cc} C_{a}\left(n\right) & =C_{a1}\left(P_{n+1}\right)-C_{a1}\left(P_{n}\right). \end{array}$$

**Proof.** See Appendix B. □

#### 3.2.2. The Average Achievable Secrecy Rate for $x_{2}$

The AASR for $x_{2}$ is calculated by
(31)$$\begin{array}{cc} \; & {\bar{C}}_{\sec}^{x_{2}}=\frac{2\tau_{0}}{T}E\left\{E\left\{\min\left\{\begin{array}{c} C_{R}^{x_{2}},\\ {\left[C^{x_{2}}-C_{{}_{E}}^{x_{2}}\right]}^{+} \end{array}\right\}\left|O_{{}_{P_{R}}}\right.\right\}\right\}\\ \; & \overset{\left(e\right)}{\approx }\frac{2\tau_{0}}{T}\sum_{n=1}^{N}E\left\{\min\left\{\begin{array}{c} -\log_{4}\left(a_{1}\right),-\log_{4}\left(b_{1}\right)\\ -\log_{4}\left(1+\frac{b_{2}}{b_{1}+Z^{-1}}\right) \end{array}\right\}\right\}\mathrm{Pr}\left(O_{P_{R,n}}\right). \end{array}$$

The approximation at (e) is achieved by using (a),(b) and (d). Similarity, we can attain an upper bound for ${\bar{C}}_{\sec}^{x_{2}}$ as follows
(32)$$\begin{array}{c} {\bar{C}}_{\sec,\mathrm{up}}^{x_{2}}=\frac{2\tau_{0}}{T}\min\left\{\begin{array}{c} -\log_{4}\left(a_{1}\right),-\log_{4}\left(b_{1}\right)-\sum_{n=1}^{N}\underset{{\bar{C}}_{E}^{x_{2}}\left(n\right)}{\underset{︸}{E\left\{\log_{4}\left(1+\frac{b_{2}}{b_{1}+\frac{1}{Z}}\right)\right\}}}\mathrm{Pr}\left(O_{{}_{P_{R},n}}\right) \end{array}\right\}. \end{array}$$

**Proposition** **5.**
*The analytical expression for ${\bar{C}}_{\sec,\mathrm{up}}^{x_{2}}$ is given by the following equation.*
(33)$${\bar{C}}_{\sec,\mathrm{up}}^{x_{2}}=\frac{2\tau_{0}}{T}\min\left\{-\log_{4}\left(a_{1}\right),-\log_{4}\left(b_{1}\right)-{\bar{C}}_{E}^{x_{2}}\right\}.$$
*where ${\bar{C}}_{\sec,\mathrm{up}}^{x_{2}}$ is calculated by*
(34)$$\begin{array}{cc} {\bar{C}}_{E}^{x_{2}} & =\sum_{n=1}^{N}\frac{\mu_{5}}{\mu_{5}b_{1}-1}\left(\mu_{10}\log_{4}\left(b_{1}\mu_{5}\right)+\mu_{11}\log_{4}\left(b_{1}\right)\right)\left(F_{{}_{X}}\left(P_{n+1}\right)-F_{{}_{X}}\left(P_{n}\right)\right). \end{array}$$


**Proof.** See Appendix C. □

For ${\bar{C}}_{\sec,\mathrm{up}}^{x_{1}}$, it is seen from the definition of $O_{\hat{Z}}$ and (A17) that the increase in the jamming signal strength at *E* leads to the increase in $\mathrm{Pr}(O_{\hat{Z}})$ and $C_{b}\left(n\right)$ (by degrading $C_{E}^{x_{1}}$), and the increase in the jamming signal strength at *R* leads to the increase in $C_{b}\left(n\right)$ (by enhancing $C_{U_{1}}^{x_{1}}$). For ${\bar{C}}_{\sec,\mathrm{up}}^{x_{2}}$, there is an upper limit for ${\bar{C}}_{\sec,\mathrm{up}}^{x_{2}}$, i.e., $R_{\mathrm{limit}}^{x_{2}}=-\log_{4}\left(\max\left(a_{1},b_{1}\right)\right)$ (as observed in (Equation 33)), and ${\bar{C}}_{\sec,\mathrm{up}}^{x_{2}}$ reaches $R_{\mathrm{limit}}^{x_{2}}$ iff $b_{1}\ll a_{1}$ or ${\bar{C}}_{E}^{x_{2}}\left(n\right)$ is very small. With the presence of jammer, the eavesdropping rate can be reduced, hence, $a_{1}$ and $b_{1}$ can be set at the same value to gain the high AASRs for both users.

### 3.3. The Average Stored Energy (ASE)

After extracting the amount of energy for information forwarding, the average remaining energy at *R* for energy storing is given by
(35)$$\begin{array}{c} \mathrm{ASE}=\tau_{0}\int_{0}^{\infty }\Phi_{{}_{\mathrm{EH}}}\left(\left(1-\theta \right)x\right)f_{{}_{X}}\left(x\right)dx-\tau_{0}\sum_{n=1}^{N}P_{{}_{R,n}}\int_{P_{n}}^{P_{n+1}}f_{{}_{X}}\left(x\right)dx. \end{array}$$

**Proposition** **6.***The analytical expression for the* ASE *is given by the following equation.*
(36)$$\mathrm{ASE}=\tau_{0}{\bar{P}}_{R,\mathrm{EH}}^{\mathrm{Total}}-\tau_{0}\sum_{n=1}^{N}P_{{}_{R,n}}\left(F_{{}_{X}}\left(P_{n+1}\right)-F_{{}_{X}}\left(P_{n}\right)\right),$$
*where ${\bar{P}}_{R,\mathrm{EH}}^{\mathrm{Total}}=\int_{0}^{\infty }\Phi_{{}_{\mathrm{EH}}}\left(\left(1-\theta \right)x\right)f_{{}_{X}}\left(x\right)dx$, for the case of $\mu_{3}\neq 0$,*
(37)$$\begin{array}{cc} {\bar{P}}_{R,\mathrm{EH}}^{\mathrm{Total}} & =P_{\mathrm{EH}}^{\max}+\frac{\lambda_{{}_{SR}}\lambda_{{}_{JR}}\left(1+e^{a_{{}_{\mathrm{EH}}}b_{{}_{\mathrm{EH}}}}\right)P_{\mathrm{EH}}^{\max}}{\mu_{3}}\left(\begin{array}{c} I_{1}\left(\frac{\lambda_{{}_{SR}}}{a_{{}_{\mathrm{EH}}}\left(1-\theta \right)P_{{}_{S}}},a_{{}_{\mathrm{EH}}}\left(1-\theta \right),e^{a_{{}_{\mathrm{EH}}}b_{{}_{\mathrm{EH}}}}\right)\\ -I_{1}\left(\frac{\lambda_{{}_{JR}}}{a_{{}_{\mathrm{EH}}}\left(1-\theta \right)P_{{}_{J}}},a_{{}_{\mathrm{EH}}}\left(1-\theta \right),e^{a_{{}_{\mathrm{EH}}}b_{{}_{\mathrm{EH}}}}\right) \end{array}\right), \end{array}$$
*and for the case of $\mu_{3}=0$,*
(38)$$\begin{array}{cc} {\bar{P}}_{R,\mathrm{EH}}^{\mathrm{Total}} & =P_{\mathrm{EH}}^{\max}-\frac{\lambda_{{}_{SR}}^{2}P_{\mathrm{EH}}^{\max}\left(1+e^{a_{{}_{\mathrm{EH}}}b_{{}_{\mathrm{EH}}}}\right)}{P_{{}_{S}}^{2}}I_{2}\left(\frac{\lambda_{{}_{SR}}}{a_{{}_{\mathrm{EH}}}\left(1-\theta \right)P_{{}_{S}}},a_{{}_{\mathrm{EH}}}\left(1-\theta \right),e^{a_{{}_{\mathrm{EH}}}b_{{}_{\mathrm{EH}}}}\right), \end{array}$$
*with $I_{1}\left(k,p,c\right)$ and $I_{2}\left(k,p,c\right)$ defined as*
(39)$$\begin{array}{c} I_{1}\left(k,p,c\right)=\frac{\Gamma (k+1)}{p\Gamma (k+2)}{}_{2}F_{1}\left(1,k+1;k+2;-c\right), \end{array}$$
(40)$$\begin{array}{c} I_{2}\left(k,p,c\right)=\frac{{}_{3}F_{2}\left(k+1,1,k+1;k+2,k+2;-c\right)}{p^{2}{\left(k+1\right)}^{2}} \end{array}$$

**Proof.** See Appendix D. □

The first element of the right-hand side (RHS) of (Equation 36), i.e., $\tau_{0}{\bar{P}}_{R,\mathrm{EH}}^{\mathrm{Total}}$, is the average harvested energy of *R*, and it is a constant for a given system configuration. The remain of the RHS of (Equation 36) is the portion of harvested energy employed for information forwarding. Therefore, the ASE depends on the selected MTPLs in OPS. To enhance ${\mathrm{SOP}}_{\mathrm{All}}$ and AASR, *R* should store less energy and use higher portion of harvested energy to assist the communication. This means that the OPS should include more MTPLs with small *“power gap”*. To guarantee a target values of ASE, the OPS should include fewer MTPLs with equal *“power gaps”*. As shown Figure 2d, the equal *“power gaps”* can allow *R* to send information at more effective power levels. For the HtT scheme, the ASE is equal to 0. Since the ASE depends on $\tau_{0}$, it receives lower values when the system spends more time for the setup phase.

## 4. Results and Discussions

In this section, we present numerical results to validate the analytical expressions presented in Section 3. Unless otherwise specified, we set $P_{{}_{S}}=P_{{}_{D}}=30$ dBm, $a_{1}=b_{1}=0.1$, $R_{\mathrm{th}}^{x_{1}}=1.1$ bits/sec/Hz, $R_{\mathrm{th}}^{x_{2}}=1$ bits/sec/Hz, $\alpha =2.3,\theta =0.5,P_{\mathrm{EH}}^{\max}=24$ mW, $a_{{}_{\mathrm{EH}}}=150,b_{{}_{\mathrm{EH}}}=0.014$ and $N_{0}=1$ (We use the parameters of the NLEH model given in [27]). We assume that $\tau_{{}_{\mathrm{setup}}}\ll T$ for the two-user case; hence, we do not examine the effects of $\tau_{{}_{\mathrm{setup}}}$ on the AASR results. The coordinates (in meters) in the two-dimensional plane of $S,R,U_{1},U_{2},E$ and *J* (as illustrated in Figure 1a) are set at $S\left(-6,1.5\right),R\left(0,0\right),U_{1}\left(6,1.5\right),U_{2}\left(9,1.5\right),E\left(1.5,-6\right)$ and J(4.5,−3), respectively. We consider the SaT scheme in two scenarios, MTPL and STPL. The STPL is a simpler case of the MTPL where *R* supports only one transmit-power level. For the MTPL, we examined four OPSs given as $O_{R,\approx \mathrm{HtT}}=\left\{0,0.2,\ldots ,23.8,P_{\mathrm{EH}}^{\max}\right\}$ mW, $O_{R,\mathrm{OPS}}^{\left(a\right)}=\left\{0,5,10,15,20,P_{\mathrm{EH}}^{\max}\right\}$ mW, $O_{R,\mathrm{OPS}}^{\left(b\right)}=\left\{0,10,15,20,P_{\mathrm{EH}}^{\max}\right\}$ mW and $O_{R,\mathrm{OPS}}^{\left(c\right)}=\left\{0,15,20,P_{\mathrm{EH}}^{\max}\right\}$ mW. For the STPL, the supported transmit-power level of *R* is examined in four cases, i.e., 5 mW, 10 mW, 15 mW, and 20 mW. Moreover, we consider the case that *R* uses the HtT scheme to assist the communication and show that the SaT scheme using $O_{R,\approx \mathrm{HtT}}$ approximates well the HtT scheme.

In Figure 3, we show the effects of required secrecy rates, $R_{\mathrm{th}}^{x_{1}}$ and $R_{\mathrm{th}}^{x_{2}}$, on the SOP. ${\mathrm{SOP}}_{\mathrm{All}}$ and ${\mathrm{SOP}}_{\mathrm{ON}}$ show increasing trends with the increases in $R_{\mathrm{th}}^{x_{1}}$ and/or $R_{\mathrm{th}}^{x_{2}}$. It is easy to show that $R_{\mathrm{limit}}^{x_{2}}=-0.5\log_{2}\left(\max\left(a_{1},b_{1}\right)\right)=1.66$ bits/s/Hz. The analytical results match well with the simulation results except for the case that $R_{\mathrm{th}}^{x_{2}}$ gets close to $R_{\mathrm{limit}}^{x_{2}}$ and $P_{{}_{R}}$ gets relatively low values. At higher values of $P_{{}_{R}}$, i.e., 15 mW and 20 mW, the accuracy of the analytical results is improved significantly. The reason is the approximations at (Equation 13), (14) and (A13) become less accurate when $P_{{}_{R}}$ is low and $R_{\mathrm{th}}^{x_{2}}$ approximates $R_{\mathrm{limit}}^{x_{2}}$, respectively. It is seen in Figure 3a that there is a change in the effective values of $P_{{}_{R}}$, which yields the lowest ${\mathrm{SOP}}_{\mathrm{All}}$ for the STPL. With the increase in $R_{\mathrm{th}}^{x_{1}}$, *R* chooses higher transmit-power levels to achieve the lowest ${\mathrm{SOP}}_{\mathrm{All}}$. This disadvantage is overcome by the MTPL. By sending information at diverse transmits powers, the MTPL shows its superiority in improving the secrecy outage performance. The HtT scheme achieves the best performance and the analytical results for the SaT scheme using $O_{R,\approx \mathrm{HtT}}$ approximate well with the simulation results for the HtT scheme.

In Figure 4, we show the effects of NOMA power-allocation factors, $\beta =a_{1}=b_{1}$, on the secrecy performance. Similarly, for given value of $R_{\mathrm{th}}^{x_{2}}$, there is an upper limit for β, denoted as $\beta_{\max}$ (using (7), (Equation 13) and (14), we can show that $\max\left\{a_{1},b_{1}\right\}\leq \beta_{\max}=2^{-2R_{\mathrm{th}}^{x2}}$). The SOP is equal to zero as β=0 or $\beta >\beta_{\max}$, and achieves good values as β is around $\frac{1}{2}\beta_{\max}$. The less accuracy of the analytical results when β approximates $\beta_{\max}$ and $P_{{}_{R}}$ gets relatively low values can be explained in a similar way as discussed in Figure 3. For the AASR results, the secrecy rates of $U_{1}$ and $U_{2}$ follow different trends, i.e., increasing function and decreasing function of β, respectively, that are the common trends for all NOMA systems. The AASR of $U_{1}$ for the MTPL outperforms that for the STPL due to the great advantage in using multiple transmit-power levels. The analytical results of the AASR of $U_{2}$ for both STPL and MTPL are the same; however, their simulation results indicate that the MTPL gives higher secrecy rates and fits the analytical results better than the STPL.

In Figure 5, we present the effect of the energy harvesting coefficient θ on the secrecy performance. As shown that the secrecy performance improves when θ increases from 0 to an optimal value of θ, denoted by $\theta^{*}$, then it degrades with further increase in θ. These trends are explained using the effect of θ on the signal strength employed for information decoding and energy harvesting (i.e., $\theta y_{R}$ and $\left(1-\theta \right)y_{R}$, respectively). As shown in Figure 5, $\theta^{*}$ varies in the range of [0.1,0.2]. The trend for ${\mathrm{SOP}}_{\mathrm{ON}}$ is similar to that for ${\mathrm{SOP}}_{\mathrm{All}}$ except for the case θ shifts to 1. Because the ${\mathrm{SOP}}_{\mathrm{ON}}$ is calculated for the case that *R* harvests enough energy to send information at power $P_{{}_{R}}⩾P_{R,1}$; hence, if $P_{R,1}\gg 0$ mW, ${\mathrm{SOP}}_{\mathrm{ON}}$ does not tend to 0 when θ shifts to 1. Moreover, the probability that *R* is ON becomes very small when θ is close to 1; hence, this causes the less accuracy in simulation results. For the $O_{R,\mathrm{HtT}}$, since *R* almost assists the communication during all block times, ${\mathrm{SOP}}_{\mathrm{ON}}$ and ${\mathrm{SOP}}_{\mathrm{All}}$ get similar values. The effect of θ observed in Figure 5 agrees with previous studies in SWIPT relaying system. The obtained results indicate that NLEH relay uses higher portions (i.e., lower values of θ) of its received signal strength for energy harvesting to archive the optimal performance as compared to studies with the LEH model (e.g., $\theta^{*}\in \left(0.23,0.38\right)$ in [4] and $\theta^{*}\in \left(0.5,0.6\right)$ in [28]).

In Figure 6, we show the effect of jammer’s location on the secrecy performance. When *J* moves far from *R*, the decrease in received power of *R* and the increase in wiretapping rate of *E* cause the lower secrecy performance. The ${\mathrm{SOP}}_{\mathrm{All}}$ and AASR converge to low positive values when *J* shifts to the coordinate (9,−10).This result indicates that without jammer, the system can satisfy low security-level requirements, such as $0.6<{\mathrm{SOP}}_{\mathrm{All}}<0.93$. In this case, the secrecy capacity only depends on the relay’s active-state probability and the difference in channel gains between legitimate links (i.e., $R\to U_{1}$ and $R\to U_{2}$) and illegitimate link (i.e., R→E). Considering the ${\mathrm{SOP}}_{\mathrm{ON}}$ of the STPL, the ${\mathrm{SOP}}_{\mathrm{ON}}$ improves during range $x_{{}_{J}}\in \left(1,2\right)$, then it degrades with the further increase in $x_{{}_{J}}$. The reason is when *J* moves toward the coordinate (3,-4), the shorter J→E distance causes the higher interference at *E* (note that the signal strength at *E* is constant due to fixed values of $P_{R}$ in the STPL) leading to the enhancement in secrecy rate. This trend is unclear in Figure 6b because the J→E distance can also affect the values of $P_{R}$. The results in Figure 6 confirm that the jamming signal plays an important role in improving the security rates.

In Figure 7, we examine the system performance under two TPSs, at *R*, i.e., $G_{{}_{R}}^{0\mathrm{mW}}$ and $G_{{}_{R}}^{5\mathrm{mW}}$ ($G_{{}_{R}}^{5\mathrm{mW}}$ involves a practical harvester where $0<P_{\mathrm{th}}^{\mathrm{act}}<5$ mW, and $G_{{}_{R}}^{0\mathrm{mW}}$ involves an ideal harvester with $P_{\mathrm{th}}^{\mathrm{act}}=0$ mW), and different transmit-power configurations at the energy sources, i.e., 23 dBm, 27 dBm and 30 dBm schemes. The output-power levels, $P_{{}_{R,n}}$, for $G_{{}_{R}}^{P_{\mathrm{EH}}^{\min}}$ is determined by $P_{{}_{R,n}}=P_{\mathrm{EH}}^{\min}+\frac{n}{N+1}\left(P_{\mathrm{EH}}^{\max}-P_{\mathrm{EH}}^{\min}\right),n=1,\ldots ,N,$ where $P_{\mathrm{EH}}^{\min}$ is the minimum power supported at EH transceiver ($P_{\mathrm{EH}}^{\min}=0$ mW and $P_{\mathrm{EH}}^{\min}=5$ mW for $G_{{}_{R}}^{0\mathrm{mW}}$ and $G_{{}_{R}}^{5\mathrm{mW}}$, respectively). There are two effects of increasing *N* on the TPSs. The first effect is the smaller “*power gap*” between two successive power levels when *N* increases. Since *R* uses the remaining energy for energy storing after it extracts amount energy for assisting the communication, the smaller “*power gap*” causes the lower stored energy at *R*; hence, the ASE is a decreasing function of *N*. When $P_{{}_{\mathrm{EH}}}^{\min}$ lifts, the higher remaining energy attained in range $\left(0,P_{\mathrm{EH}}^{\min}\right)$ can yield the greater ASE (as shown in Figure 7, the ASE for $G_{{}_{R}}^{5\mathrm{mW}}$ is greater than that for $G_{{}_{R}}^{0\mathrm{mW}}$). The second effect is that the transmit powers of *R* are spread over $\left(P_{\mathrm{EH}}^{\min},P_{\mathrm{EH}}^{\max}\right)$, such as $P_{{}_{R,1}}$ and $P_{{}_{R,N}}$ shift to $P_{{}_{\mathrm{EH}}}^{\min}$ and $P_{{}_{\mathrm{EH}}}^{\max}$, respectively. This makes *R* help the communication more frequently with more appropriate transmit powers. Therefore, the ${\mathrm{SOP}}_{\mathrm{All}}$ and AASR achieve better values as *N* increases. The frequent operation of *R* achieved by low values of $P_{{}_{R,1}}$ does not give any advantages in enhancing ${\mathrm{SOP}}_{\mathrm{ON}}$. On the other hand, lower values of $P_{{}_{R,1}}$ causes more outage events. As a result, the ${\mathrm{SOP}}_{\mathrm{ON}}$ becomes worse as *N* increases.

For three transmit-power configurations, the higher transmit powers yield better results in ${\mathrm{SOP}}_{\mathrm{All}}$ and AASR. The ASEs of $G_{{}_{R}}^{0\mathrm{mW}}$ are increasing functions of $P_{{}_{S}}$ and $P_{{}_{J}}$. An interesting result is the ASEs of $G_{{}_{R}}^{5\mathrm{mW}}$ can achieve better values at lower values of $P_{{}_{S}}$ and $P_{{}_{J}}$. For instance, when N≥3, the 27 dBm scheme yields a higher ASE as compared to the rest schemes. This result can be explained using the relay’s received power and the “*power gap*”. The relay’s received power for the 27 dBm scheme is not too low to gain low ASEs as observed at the 23 dBm scheme (the ${\mathrm{SOP}}_{\mathrm{All}}$ verify that *R* is almost idle in the 23 dBm scheme due to too low received power), and it is not too high to lose the great advantage in harvesting energy in range (0,5) mW. These results provide useful insight in designing practical systems, such as choosing appropriate powers at the energy sources and the optimum transmit-power strategies at the EH transceivers to satisfy the requirements in secrecy capacity and amount of stored energy.

## 5. Conclusions

This paper studied the secrecy performance of a NOMA cooperative system via a NLEH relay. A friendly jammer, which can be an authorized node in the network, was employed to enhance the secrecy performance by rising the received power at the relay and interfering with the wiretapping link. A SaT scheme was proposed aiming to design feasible NLEH transceivers and enable energy-storing mechanism. Moreover, this scheme allowed the calculations for the ${\mathrm{SOP}}_{\mathrm{All}}$, ${\mathrm{SOP}}_{\mathrm{ON}}$ and AASR to become mathematical tractable. The analytical results obtained using the SaT scheme can enable analytical performance evaluations for the HtT-based NLEH relaying systems, which cannot be achieved by the direct approaches. The current study considers NOMA communications for two users; however, the analysis results for this study can be extended to the multiple-user scenario. The accuracy of the calculation was verified by simulation which showed that the analytical results agree well with simulation results for most considered cases. In addition, it pointed out the special cases causing the less accuracy for the analytical results, i.e., the target secrecy rate of the FU is chosen with inappropriate values or the relay’s transmit power is too low. The STPL was examined to confirm the superiority of the MTPL in enhancing the secrecy performance. Moreover, the effects of system parameters on the SOP, AASR and ASE were investigated. These effects provided insight into the system design. For instance, the OPS should include both high and low MTPLs with equal *“power gaps”* to archive the best secrecy performance, and the number of MTPLs is smaller enough to allow energy storing at relay; for a given OPS, the transmit powers at the source and jammer could be appropriately selected to guarantee the ASE at relay; the use of jammer produces a great advantages in enhancing harvested energy and secrecy performance; and the analytical results allow network designers to find optimal values for key system parameters, e.g., the optimal power-splitting ratio for the considered system is around 0.2. The more general model where the eavesdropper is not close to the cell edge and is capable of receiving signals from both the source and relay will be our future works. To confine the eavesdropping capability for this model, different transmit codewords can be exploited [29].

## Figures and Tables

**Figure 1 sensors-20-01047-f001:**
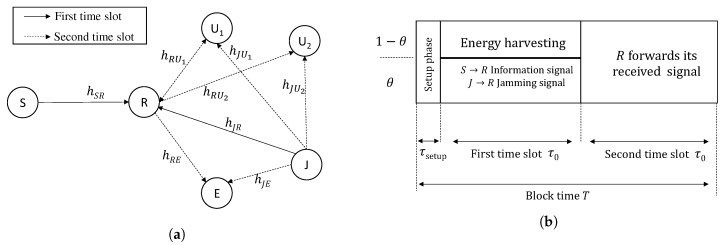
(**a**) The system model and (**b**) The power-splitting policy at *R*.

**Figure 2 sensors-20-01047-f002:**
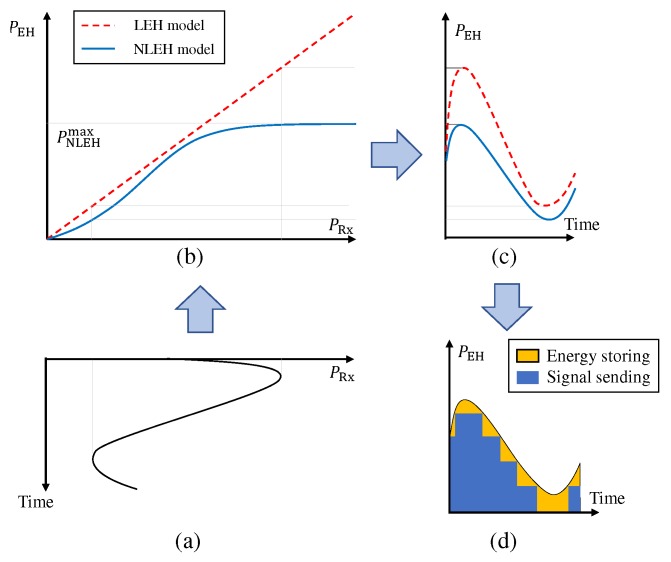
(**a**) The received power at the harvester, (**b**) the LEH and NLEH models, (**c**) the harvested energy and (**d**) the SaT scheme with NLEH.

**Figure 3 sensors-20-01047-f003:**
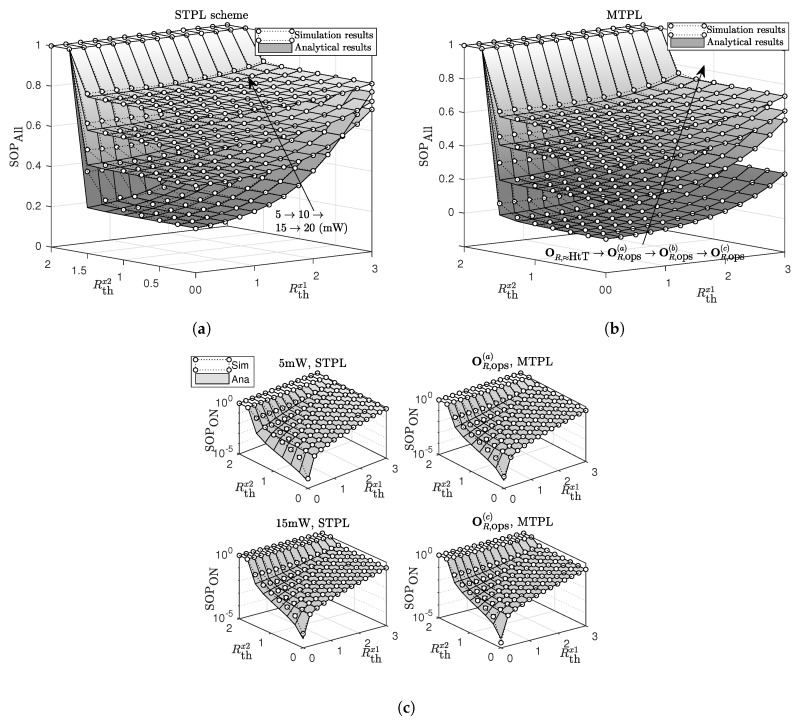
The effect of $R_{\mathrm{th}}^{x_{1}}$ and $R_{\mathrm{th}}^{x_{2}}$ on (**a**) SOP_All_ for the STPL scheme, (**b**) SOP_All_ for the MTPL scheme, and (**c**) SOP_ON_ for both STPL and MTPL schemes.

**Figure 4 sensors-20-01047-f004:**
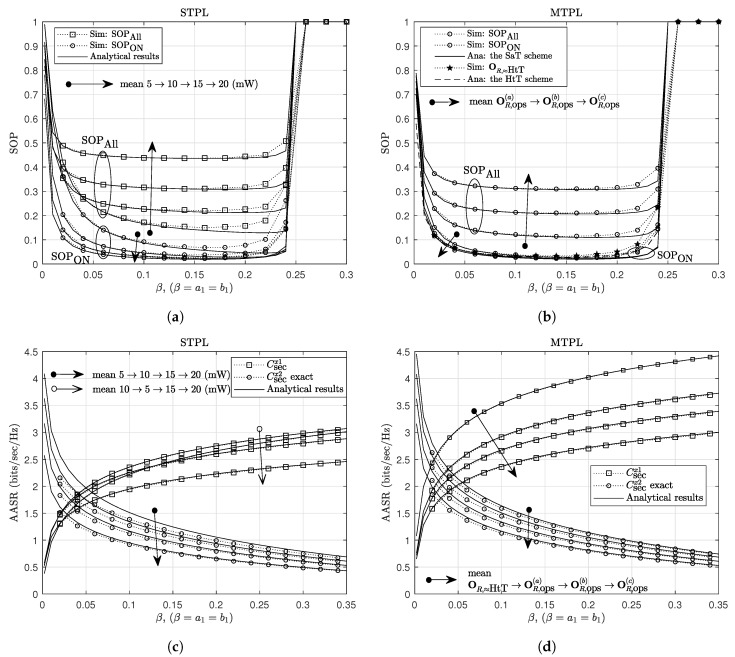
The effect of NOMA power-allocation factors on (**a**) the SOP for the STPL scheme, (**b**) the SOP for the MTPL scheme, (**c**) the AASR for the STPL scheme, and (**d**) the AASR for the MTPL scheme. Other parameters: **J**(1.5,−3).

**Figure 5 sensors-20-01047-f005:**
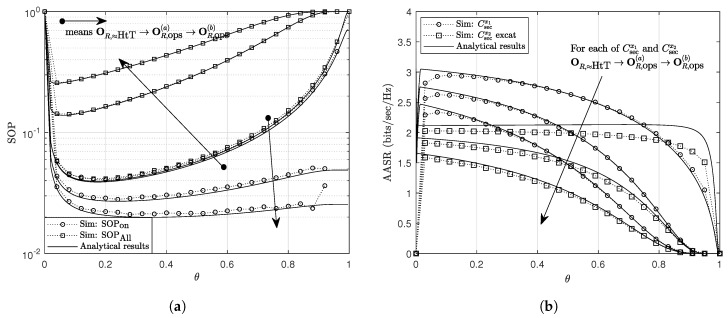
The effect of *θ* on (**a**) the SOP and (**b**) the AASR. Other parameters: *P_S_* = *P_J_* = 27 dBm, *a*_1_ = *b*_1_ = 0.1, ($R_{\mathrm{th}}^{x_{1}}$, $R_{\mathrm{th}}^{x_{2}}$) = (1, 0.6) bits/sec/Hz.

**Figure 6 sensors-20-01047-f006:**
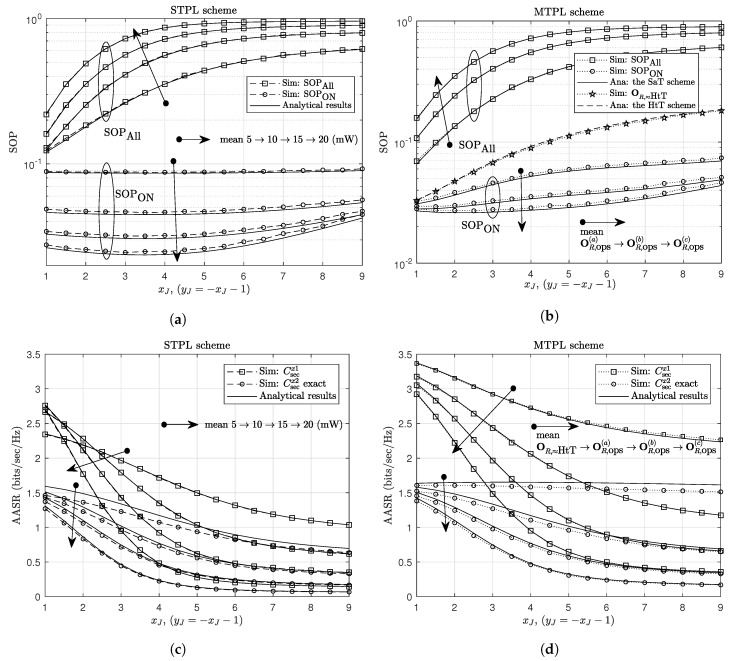
The effect of the jammer’s location on (**a**) the SOP for the STPL scheme, (**b**) the SOP for the MTPL scheme, (**c**) the AASR for the STPL scheme, and (**d**) the AASR for the MTPL scheme.

**Figure 7 sensors-20-01047-f007:**
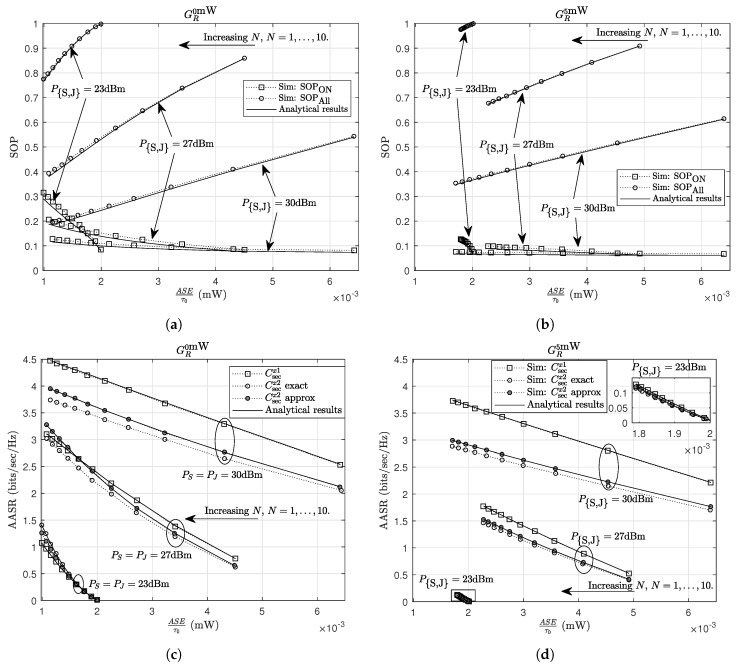
The trade-off between the SOP and the ASE for (**a**) $G_{{}_{R}}^{0\mathrm{mW}}$ and (**b**) $G_{{}_{R}}^{5\mathrm{mW}}$; and the trade-off between the AASR and the ASE for (**c**) $G_{{}_{R}}^{0\mathrm{mW}}$ and (**d**) $G_{{}_{R}}^{5\mathrm{mW}}$.

**Table 1 sensors-20-01047-t001:** The list of parameters $\mu_{k},1\leq k\leq 11$.

Parameters $\mu_{1}$–$\mu_{4}$	Parameters $\mu_{5}$–$\mu_{8}$	Parameters $\mu_{9}$–$\mu_{11}$
$\mu_{1}=\frac{2a_{1}\theta P_{s}}{\left(1+\theta \right)N_{0}}$	$\mu_{5}=\frac{\lambda_{{}_{JE}}P_{R,n}}{\lambda_{{}_{RE}}P_{J}}$	$\mu_{9}=\frac{\lambda_{{}_{RU_{1}}}2^{2R_{\mathrm{th}}^{x_{1}}}N_{0}}{P_{R}}$
$\mu_{2}=\frac{2a_{2}\theta P_{s}}{\left(1+\theta \right)N_{0}}$	$\mu_{6}=\frac{2^{-2R_{\mathrm{th}}^{x_{2}}}}{b_{1}}-1,\mu_{6}\in \left(-1,\frac{b_{2}}{b_{1}}\right]$	$\mu_{10}=\frac{-b_{2}}{\mu_{2}-1}$
$\mu_{3}=\lambda_{{}_{SR}}P_{J}-\lambda_{{}_{JR}}P_{S}$	$\mu_{7}=\frac{1}{\frac{b_{2}}{\mu_{6}}-b_{1}}=\frac{2^{-2R_{\mathrm{th}}^{x_{2}}}-b_{1}}{b_{1}\left(1-2^{-2R_{\mathrm{th}}^{x_{2}}}\right)}$	$\mu_{11}=\frac{1-\mu_{2}b_{1}}{\mu_{2}-1}$
$\mu_{4}=\frac{2^{2R_{\mathrm{th}}^{x_{2}}}-1}{\mu_{1}}\max\left\{1,\frac{a_{1}}{1-a_{1}2^{2R_{\mathrm{th}}^{x_{2}}}}\right\}$	$\mu_{8}=\exp\left(\frac{\lambda_{{}_{RU_{1}}}N_{0}\left(1-2^{2R_{\mathrm{th}}^{x_{1}}}\right)}{b_{1}P_{R}}\right)$	

**Table 2 sensors-20-01047-t002:** The analytical expressions for $P_{b}\left(n\right),P_{c}\left(n\right)$ and $P_{d}^{\approx }\left(n\right)$

Functions	Analytical Expressions	Cases
$P_{b}\left(n\right)=$	$\begin{array}{c} F_{{}_{X}}\left(P_{n+1}\right)-F_{{}_{X}}\left(P_{n}\right)-P_{c}\left(n\right) \end{array}.$	
$P_{c}\left(n\right)=\left\{\begin{array}{ll} 0, & \left\{1\leq a_{1}2^{2R_{\mathrm{th}}^{x_{2}}}\right\}\cup \left\{\begin{array}{c} 1>a_{1}2^{2R_{\mathrm{th}}^{x_{2}}},\\ P_{n+1}\leq \mu_{4}P_{{}_{S}} \end{array}\right\}.\\ \begin{array}{c} \exp\left(-\lambda_{{}_{SR}}\mu_{4}\right)+\frac{\lambda_{{}_{JR}}P_{{}_{S}}}{\mu_{3}}\exp\left(-\frac{\lambda_{{}_{SR}}P_{n+1}}{P_{{}_{S}}}\right)\\ \;-\frac{\lambda_{{}_{SR}}P_{J}}{\mu_{3}}\exp\left(-\frac{\lambda_{{}_{JR}}P_{n+1}+\mu_{4}\mu_{3}}{P_{J}}\right), \end{array} & \left\{\begin{array}{c} 1>a_{1}2^{2R_{\mathrm{th}}^{x_{2}}},\\ P_{n}\leq \mu_{4}P_{{}_{S}}<P_{n+1},\mu_{3}\neq 0 \end{array}\right\}.\\ \begin{array}{c} \exp\left(-\lambda_{{}_{SR}}\mu_{4}\right)-\lambda_{{}_{SR}}\frac{P_{n+1}-\mu_{4}P_{s}}{P_{J}}\exp\left(-\frac{\lambda_{{}_{SR}}P_{n+1}}{P_{{}_{S}}}\right)\\ \;-\exp\left(-\frac{\lambda_{{}_{JR}}P_{n+1}+\mu_{4}\mu_{3}}{P_{J}}\right), \end{array} & \left\{\begin{array}{c} 1>a_{1}2^{2R_{\mathrm{th}}^{x_{2}}},\\ P_{n}\leq \mu_{4}P_{{}_{S}}<P_{n+1},\mu_{3}=0 \end{array}\right\}.\\ \begin{array}{c} \frac{\lambda_{{}_{SR}}P_{J}}{\mu_{3}}\exp\left(-\frac{\mu_{4}\mu_{3}}{P_{J}}\right)\left(\exp\left(-\frac{\lambda_{{}_{JR}}P_{n}}{P_{J}}\right)-\exp\left(-\frac{\lambda_{4}P_{n+1}}{P_{J}}\right)\right)\\ \;+\frac{\lambda_{{}_{JR}}P_{{}_{S}}}{\mu_{3}}\left(\exp\left(-\frac{\lambda_{{}_{SR}}P_{n+1}}{P_{{}_{S}}}\right)-\exp\left(-\frac{\lambda_{{}_{SR}}P_{n}}{P_{{}_{S}}}\right)\right), \end{array} & \left\{\begin{array}{c} 1>a_{1}2^{2R_{\mathrm{th}}^{x_{2}}},\\ \mu_{4}P_{{}_{S}}<P_{n},\mu_{3}\neq 0 \end{array}\right\}.\\ \begin{array}{c} \exp\left(-\frac{\mu_{4}\mu_{3}}{P_{J}}\right)\left(\exp\left(-\frac{\lambda_{{}_{JR}}P_{,n}}{P_{J}}\right)-\exp\left(-\frac{\lambda_{{}_{JR}}P_{n+1}}{P_{J}}\right)\right)\\ \;+\lambda_{{}_{JR}}\exp\left(-\frac{\lambda_{{}_{SR}}P_{n}}{P_{{}_{S}}}\right)\frac{P_{n}-\mu_{4}P_{{}_{S}}}{P_{J}}-\lambda_{{}_{JR}}\exp\left(-\frac{\lambda_{{}_{SR}}P_{n+1}}{P_{s}}\right)\frac{P_{n+1}-\mu_{4}P_{s}}{P_{J}}, \end{array} & \left\{\begin{array}{c} 1>a_{1}2^{2R_{\mathrm{th}}^{x_{2}}},\\ \mu_{4}P_{{}_{S}}<P_{n},\mu_{3}=0 \end{array}\right\}. \end{array}\right.$		
$P_{d}^{\approx }\left(n\right)=\left\{\begin{array}{ll} 1, & \mu_{6}\in \left(-1,0\right].\\ \begin{array}{c} 1-\mu_{8}+\frac{\mu_{5}\mu_{8}\exp\left(-\mu_{7}\mu_{9}\right)}{\mu_{7}+\mu_{5}}+\mu_{5}\mu_{8}\mu_{9}\exp\left(\mu_{5}\mu_{9}\right)\\ \;\times \left(Ei\left(-\mu_{7}\mu_{9}-\mu_{5}\mu_{9}\right)-Ei\left(-\mu_{5}\mu_{9}\right)\right), \end{array} & \mu_{6}\in \left(0,\frac{b_{2}}{b_{1}}\right). \end{array}\right.$		

**Table 3 sensors-20-01047-t003:** The analytical expressions for $C_{b}\left(n\right)$ and $C_{a1}\left(P\right)$.

Functions	Analytical Expressions	Cases
$C_{b}\left(n\right)=$	$\frac{1}{\ln\left(4\right)\left(\mu_{5}b_{1}-1\right)}\left(\exp\left(\frac{\lambda_{{}_{RU_{1}}}N_{0}}{b_{1}P_{R,n}}\right)Ei\left(-\frac{\lambda_{{}_{RU_{1}}}N_{0}}{b_{1}P_{R,n}}\right)-b_{1}\mu_{5}\exp\left(\frac{\mu_{5}\lambda_{{}_{RU_{1}}}N_{0}}{P_{R,n}}\right)Ei\left(-\frac{\mu_{5}\lambda_{{}_{RU_{1}}}N_{0}}{P_{R,n}}\right)\right)$.	
$C_{a1}\left(P\right)=\left\{\begin{array}{ll} \frac{1}{\ln(4)}\left(\begin{array}{c} -\exp\left(\frac{\lambda_{{}_{SR}}}{\mu_{1}}\right)Ei\left(-\frac{\lambda_{{}_{SR}}}{\mu_{1}}\right)+\exp\left(\frac{\lambda_{{}_{SR}}}{\mu_{1}}\right)Ei\left(-\frac{\lambda_{{}_{SR}}P}{P_{{}_{S}}}-\frac{\lambda_{{}_{SR}}}{\mu_{1}}\right)\\ +\frac{\lambda_{{}_{JR}}P_{{}_{S}}}{\mu_{3}}\exp\left(-\frac{\lambda_{{}_{SR}}P}{P_{s}}\right)\ln\left(1+\frac{\mu_{1}P}{P_{s}}\right)\\ +\frac{\lambda_{{}_{SR}}P_{J}}{\mu_{3}}\exp\left(\frac{\mu_{3}}{\mu_{1}P_{J}}-\frac{\lambda_{{}_{JR}}P}{P_{J}}\right)\left(Ei\left(-\frac{\mu_{3}}{\mu_{1}P_{J}}\right)-Ei\left(-\frac{\mu_{3}}{P_{J}}\left(\frac{P}{P_{{}_{S}}}+\frac{1}{\mu_{1}}\right)\right)\right) \end{array}\right), & \mu_{3}\neq 0.\\ \frac{1}{\ln(4)}\left(\begin{array}{c} \exp\left(\frac{\lambda_{{}_{SR}}}{\mu_{1}}\right)Ei\left(-\frac{\lambda_{{}_{SR}}P}{P_{{}_{S}}}-\frac{\lambda_{{}_{SR}}}{\mu_{1}}\right)-\exp\left(\frac{\lambda_{{}_{SR}}}{\mu_{1}}\right)Ei\left(-\frac{\lambda_{{}_{SR}}}{\mu_{1}}\right)+\frac{\lambda_{{}_{SR}}\exp\left(-\frac{\lambda_{{}_{SR}}P}{P_{s}}\right)}{P_{S}}P\\ -\left(\frac{\lambda_{{}_{SR}}}{\mu_{1}}+\frac{\lambda_{{}_{SR}}P}{P_{S}}+1\right)\exp\left(-\frac{\lambda_{{}_{SR}}P}{P_{s}}\right)\ln\left(1+\frac{\mu_{1}P}{P_{{}_{S}}}\right) \end{array}\right), & \mu_{3}=0. \end{array}\right.$		

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
