# Peer review of "Secure Communication in Cooperative SWIPT NOMA Systems with Non-Linear Energy Harvesting and Friendly Jamming"

_sensors, 2020, doi:10.3390/s20041047_

Round 1

Reviewer 1 Report

Providing a new mathematical formula for calculating various performance indicators is beneficial in itself. 
However, the benefits of providing these equations do not clearly reveal the contributions to design and interpretation of the system.
For example, the last sentence of the Conclusion does not give specific results on the technical intuition of system design. 

Author Response

Dear Reviewer,

Thank you for your valuable comments for improving the submitted manuscript.

Please, refer to attached file. Thank you.

Reviewer 2 Report

This paper studies secure communications in a cooperative SWIPT NOMA system with non-linear energy harvesting, and a friendly jammer exists to send jamming signals in order to protect the transmission from eavesdropping. The authors employ an SaT scheme including MTPLs, and derive closed-form expressions for different metrics, such as SOP, ASR, and ASE, respectively. Overall, PLS in SWIPT-based NOMA systems has already been investigated by existing literature with an LEH model, and the novelty and theoretical contribution of the current work considering an NLEH model is somewhat limited. Besides, the following issues should be well dealt with before it can be accepted for publication.

The system model needs more explanations. 1) the CSI-based key generation process between legitimate users should be done every transmission block, and it maybe more appropriate to take the consumed time into consideration. 2) By assuming that there is no direct link from S to E, this paper implies that the eavesdropper’s location is known in advance. This should be justified since we generally discuss a passive eavesdropper whose position knowledge cannot be obtained easily. As an alternative, the authors are more encouraged to consider the direct link from S to E and use specific secure DF relaying protocols to protect the secure transmission just as done in the following reference [R1]. 3) It should be justified that why the authors evaluate SOP and ASR for a single transmission scheme. In fact, the SOP is for fixed transmission rates while the ASR is for variable transmission rates.

[R1] Tong-Xing Zheng, Hui-Ming Wang, Feng Liu, and Moon Ho Lee, “Outage constrained secrecy throughput maximization for DF relay networks,” IEEE Transactions on Communications, vol. 63, no. 5, pp. 1741-1755, May 2015.a

The novelty and technical contribution of the current work should be elaborated. For example, the significance of considering the NLEH model should be emphasized and maybe the difficulty should be pointed out. Although various closed-form expressions have been derived for the metrics of interest, they are too unwieldy to analyze, and the paper lacks theoretical analyses and insights rather than numerical results. The authors are highly encouraged to do some in-depth analyses based on the obtained expressions to provide useful insights into practical designs. The presentation of this paper should be improved. For example, it is not quite convenient for readers to connect $h_{k}$ for $k=0,;;;,7$ to the corresponding channels. It is better to use $h_{I,j}$ to denote the channel from node $i$ to node $j$ as (12). There are too many notations/symbols which makes the paper not easy to follow. Some figures are not easy to read, e.g., Fig. 3-b. In (24) and (25), it would be better if the authors specify the random variables over which each expectation is operated. The authors are suggested to give a diagram to show the coordinates of the nodes in the simulation part. Some typos and grammar issues should be fixed, e.g., “Fig. 2a-c shows”, “The randomness of PRx effects”, “As show that”, “it degrades with further increases in”, “is closed to 1”, “The results in Fig. 6 confirms that”, “The first effect is the ‘power gap’ between two successive power levels become smaller.”, etc.

Author Response

(The authors gave the same response as above.)

Reviewer 3 Report

This paper studies the secure communication of a relaying system with the NOMA technology. The authors consider the SaT model and analyze the performance like average stored energy, secrecy rate, secrecy outage probability. To validate the performance, they also conduct numerical simulations and provides with simulation results to show the performance of the proposed scheme. This paper considers a widely-studied problem, and the paper is readable. The reviewer has the following major concerns.

1. The paper considers secure communication in wireless communication systems. However, the definition of secure communication is not clear.

2. It is not clear what kind of scheme is proposed to achieve secure communication. Please add more details in the revised version.

3. The "Store-and-Transmit" has been proposed in the literature, and has been widely studied. Thus, it compromises the novelty of this paper.

4. It would be better if the paper can provide readers with something unique when it uses SaT model.

5. Minor: There are a couple of typos and grammar issues. Please proofread the paper.

To sum up, this paper considers a widely-studied problem, and the definition is not clear. Thus, both novelty and significance are compromised.

Author Response

(The authors gave the same response as above.)

Reviewer 4 Report

This paper investigates the secure communication with NOMA and energy harvesting. The topic is timely and interesting. This paper is well written. The reviewer may have the following concerns:

Since only two users are selected to perform NOMA, it would be better to add a footnote that states the work can be extended to the general multiple user case. In the derivations, such as Eqs. (7) and (13), the approximation needs to well justified. In the simulations, it would be better to compare the results with other papers. The language of this paper can be improved. In the related work, there are also some works investigating NOMA with energy harvesting [R1-R2]. In the related work, in addition to power splitting, it would be point out time sharing for energy harvesting, as in paper Optimal fairness-aware time and power allocation in wireless powered communication networks.

[R1] Energy efficient resource allocation in machine-to-machine communications with multiple access and energy harvesting for IoT

[R2] Energy efficient resource allocation for machine-to-machine communications with NOMA and energy harvesting

Author Response

(The authors gave the same response as above.)

Round 2

Reviewer 2 Report

The authors have satisfactorily addressed all the comments from the reviewer.

Just one suggestion: please attach a one-to-one response letter with the main file next time.

Reviewer 3 Report

No new comments

Reviewer 4 Report

Thank you very much for the response. All the concerns have been well settled.